# AsymDreamer: Safe Reinforcement Learning From Pixels with Privileged World Models

## Abstract

Safe Reinforcement Learning from partial observations frequently struggles with rapid performance degradation and often fails to satisfy safety constraints. Upon deeper analysis, we attribute this problem to the lack of necessary information in partial observations and inadequate sample efficiency. World Models can help mitigate this issue, as they offer high sample efficiency and the capacity to memorize historical information. In this work, we introduce AsymDreamer, an approach based on the Dreamer framework that specializes in exploiting low-dimensional privileged information to build world models, thereby enhancing the prediction capability of critics. To ensure safety, we employ the Lagrangian method to incorporate safety constraints. Additionally, we formulate our approach as an Asymmetric CPOMDPs (ACPOMDPs) framework and analyze its superiority compared to the standard CPOMDP framework. Various experiments conducted on the Safety-Gymnasium benchmark demonstrate that our approach outperforms existing approaches dramatically in terms of performance and safety.

## 1 Introduction

As reinforcement learning (RL) has been successfully applied to various control problems Mnih et al. (2015); Yu et al. (2019), ensuring safety is crucial for real-world deployment Dulac-Arnold et al. (2021); Liu et al. (2021) . Given that partial observability is a fundamental aspect of real-world RL control problems Baisero & Amato (2021), Safe Reinforcement Learning (SafeRL) must account for partial observations. These problems are often formulated as Constrained Partially Observable Markov Decision Processes (CPOMDPs) Lee et al. (2018), where the agent operates based on a history of past observations and actions, without direct access to the true underlying states. Although significant research has been dedicated to addressing these challenges, most approaches either fail to strictly satisfy safety constraints or experience performance degradation due to insufficient critical information and low sample efficiency.

Model-based reinforcement learning (MBRL)Hafner et al. (2019); Deisenroth & Rasmussen (2011) has shown promise in overcoming these challenges by utilizing a world model that captures environmental dynamics and generates task-specific predictions from past observations and actions. This allows agents to learn from imaginary rollouts, rather than relying solely on sampled real-world trajectories LeCun & Courant (2022), which enhances both sample efficiency and safety. However, while the world model retains historical information, it does not fully resolve the issue of missing critical information. MBRL typically combines the world model with actor-critic methods for policy optimization. Unfortunately, the critic's slow or inaccurate learning of value functions can create a performance bottleneck for the policy.

Since training is often conducted in simulators, there is potential to leverage privileged information during training to reduce uncertainty from partial observations Pinto et al. (2017); Salter et al. (2021); Baisero & Amato (2021). Actor-critic methods, in particular, can handle asymmetric inputs, where the actor receives historical information and the critic accesses privileged information such as true states. This asymmetry is possible because the critic is used only during training and is not required during the agent's deployment. However, since the actor and critic share the same world model, encoding privileged information into the model may cause the actor to become dependent on it, conflicting with the requirement that the actor operates purely based on historical information during deployment.

In this work, we address the challenge of exploiting asymmetric inputs in MBRL under the CPOMDPs framework. We propose AsymDreamer, a novel algorithm that uses privileged information to construct a world model specifically for the critic. Additionally, we formulate our approach as the Asymmetric Constrained Partially Observable Markov Decision Processes (ACPOMDPs) framework and demonstrate its theoretical advantages. Our key contributions are summarized as follows:

- We introduce the ACPOMDPs framework, an extension of the CPOMDPs that allows the actor and critic to receive asymmetric inputs. We theoretically prove that asymmetric inputs reduce the number of critic updates and lead to a more optimal policy compared to standard CPOMDPs framework.

- We propose AsymDreamer, a novel MBRL approach that constructs two world models: one for the actor based on historical information, and another for the critic, which leverages privileged information.

- We integrate AsymDreamer with the Lagrangian method Nocedal & Wright (2006); Li et al. (2021), achieving competitive performance on the Safety-Gymnasium benchmark Ji et al. (2023b) and demonstrating strong adaptability to complex scenarios.

## 2 RELATED WORK

### 2.1 SAFE MODEL-BASED REINFORCEMENT LEARNING

Model-based reinforcement learning (RL) approaches Moerland et al. (2022); Polydoros & Nalpantidis (2017) present significant advantages for solving safe RL problems by facilitating the modeling of environmental dynamics. These approaches can be classified into two primary categories: planning-based methods Hafner et al. (2019) and learning-based methods Berkenkamp et al. (2017). Planning-based methods do not directly incorporate costs into the policy update process; instead, they implement an explicit planning step prior to action execution. In contrast, learning-based methods integrate costs directly into policy updates, utilizing the world model to enhance sample efficiency.

**Planning-based methods** Among planning-based methods, Koller et al. (2019); Wabersich & Zeilinger (2021); Zwane et al. (2023) enable safe action sampling through a combination of Gaussian Processes and model predictive control (MPC). Additionally, Liu et al. (2020) employ ensembles of neural networks (NN), the Cross Entropy Method (CEM) Kroese et al. (2006), and rejection sampling to optimize the expected returns of safe action sequences. Recent work by (Huang et al., 2024) has achieved zero-cost performance by integrating the constrained Cross-Entropy Method (CCEM) Wen & Topcu (2018) while considering long-term rewards and costs. Nonetheless, planning-based methods encounter challenges with myopic decisions due to the limited scope of planning and the absence of critics.

**Learning-based methods** Jayant & Bhatnagar (2022); Thomas et al. (2022) facilitate the integration of model-free algorithms with safety constraints by employing ensemble Gaussian models. Alternatively, Zanger et al. (2021) use NNs and constrained model-based policy optimization. but do not leverage model uncertainty within an optimistic-pessimistic framework. Recently, LAMBDA As et al. (2022) integrate the Bayesian methods with the Dreamer Hafner et al. (2020) framework to quantify uncertainty in the estimated model, employing the Lagrangian method to incorporate safety constraints. Similarly, Safe-SLAC Hogewind et al. (2022) integrates the Lagrangian mechanism into the SLAC framework established Lee et al. (2020) to address the problem of safe reinforcement learning from pixel observations. However, from the perspective of partially observable Markov decision processes (POMDPs) Kaelbling et al. (1998), constructing world models solely from partial observations does not fully exploit the potential of these models.

### 2.2 LEVERAGING PRIVILEGED INFORMATION

The use of asymmetric inputs is not uncommon in the single-agent domain. Pinto et al. (2017); Salter et al. (2021); Baisero & Amato (2021) utilize asymmetric actor-critic methods to accelerate the training of the critic by granting access to privileged information while providing only images to the actor. Baisero et al. (2022) introduce Asymmetric DQN, an asymmetric variant of DQN designed to

address partially observable Markov decision processes (POMDPs).Yamada et al. (2023) represent the first attempt to utilize privileged information in the training of world models. However, this method employs privileged information by distilling the learned latent dynamics model from the teacher to the student world model. Since this process of model distillation inevitably leads to a loss of information, the current exploration of world models using privileged information remains inadequate.

## 3 PRELIMINARIES

In this section, we provide a brief overview of the Constrained Partially Observable Markov Decision Processes (CPOMDPs), which is used to formulate safety constraints in sequential decision making problems under partial observations.

### 3.1 CONSTRAINED PARTIALLY OBSERVABLE MARKOV DECISION PROCESSES (CPOMDPS)

Sequential decision making problems under partial observations are typically formulated as a Partially Observable Markov Decision Processes (POMDPs) Egorov et al. (2017), represented as the tuple $(S, A, P, R, Z, O, \gamma)$. The state space is denoted as $S$ and the action space as $A$. The transition probability function $P(s'|s, a)$ captures the likelihood of the agent moving from state $s$ to state $s'$ upon taking action $a$. $Z$ is the observation space, $O(z|s', a)$ stands for the observation probability. The reward function $R : S \times A \longrightarrow \mathbb{R}$ specifies the reward obtained when transitioning from state $s$ to $s'$ via action $a$. The discount factor is represented by $\gamma$. In a Partially Observable Markov Decision Processes (POMDPs) framework, the agent has access only to the observations $z_t$ and actions $a_t$ at each time step $t$, without direct knowledge of the underlying state of the environment. As a result, the agent must maintain a belief state $b_t$, where $b_t(s) = Pr(s_t = s|h_t, b_0)$ represents the probability distribution over possible states $s$, given the history $h_t = \{z_0, a_0, z_1, a_1, \ldots, a_{t-1}, z_t\}$ of past actions and observations, and the initial belief state $b_0$. With the belief state $b$, the POMDP can be understood as the belief-state MDP $(B, A, \tau, R_B, \gamma)$ We denote the set consisting of all possible belief states as $B$, the belief reward function as $R_B(b, a) = \sum_{s \in S} b(s)R(s, a)$, the transition function as $\tau(b, a, z)$. For simplicity, we write $\tau(b, a, z)$ as $b^{a,z}$. Crucially, the agent's policy is denoted as $\pi_\theta$, which defines the probability distribution over actions $a$ given the current belief state $b$, i.e., $\pi_\theta(a \mid b)$, where $\theta$ is a learnable network parameter. The objective in a POMDP is to maximize the long-term belief expected reward $V_R(b_0)$:

$$\max_\pi V_R(b_0) = \mathbb{E}_{a_t \sim \pi}[\sum_{t=0}^{\infty} \gamma^t R_B(b_t, a_t) | b_0] \tag{1}$$

Constrained POMDPs (CPOMDPs) is a generalization of POMDPs. It is formally defined by tuple $(S, A, P, R, Z, O, \mathbb{C}, d, \gamma)$ The cost function set $\mathbb{C} = \{(C_i, b_i)\}_{i=1}^m$ comprises individual cost functions $C_i$ and their corresponding cost thresholds $b_i$. The goal is to compute an optimal policy that maximizes the long-term belief expected reward $V_R(b_0)$ while bounding the long-term belief expected costs $V_{C_i}(b_0)$:

$$\max_\pi V_R(b_0) = \mathbb{E}_{a_t \sim \pi}[\sum_{t=0}^{\infty} \gamma^t R_B(b_t, a_t) | b_0]$$
$$s.t. V_{C_i}(b_0) = \mathbb{E}_{a_t \sim \pi}[\sum_{t=0}^{\infty} \gamma^t C_{iB}(b_t, a_t) | b_0] \leq b_i, \forall i \in [m] \tag{2}$$

In practical implementations, the $V_R(b_0)$ are updated by the Bellman optimal equation:

$$V_R^*(b) = \max_{a \in A} \left[ R_B(b, a) + \gamma \sum_{z \in Z} Pr(z|b, a)V_R^*(b^{a,z}) \right] \tag{3}$$

and the $V_{C_i}(b_0)$ is updated equivalently. Consequently, the optimal policy of CPOMDPs is:

$$\pi_\star = \arg\max_{a \in A} \left[ R_B(b, a) + \gamma \sum_{z \in Z} Pr(z|b, a)V_R^*(b^{a,z}) \right]$$
$$s.t. V_{C_i}(b) \leq b_i, \forall i \in [m]$$

# 4 ASYMMETRIC CONSTRAINED PARTIALLY OBSERVABLE MARKOV DECISION PROCESSES (ACPOMDPs)

In this section, we introduce our formulation of the Asymmetric Constrained Partially Observable Markov Decision Processes (ACPOMDPs) and compare it with the standard CPOMDPs. This comparison highlights the advantages of utilizing an asymmetric architecture, particularly in terms of improving sample efficiency and achieving better policy performance under safety constraints.

## 4.1 FRAMEWORK SETUP

We propose Asymmetric Constrained Partially Observable Markov Decision Processes (ACPOMDPs), a relaxed variant of CPOMDPs. The key distinction is that ACPOMDPs assumes the availability of the underlying states when computing the long-term expected values. Our framework is grounded in the actor-critic algorithm, where the actor optimizes the policy $\pi$, while the critic estimates the long-term expected values $V_R$ and $V_C$. In contrast to the standard actor-critic algorithm, where both the actor and the critic only have access to the history $h_t = \{z_0, a_0, z_1, a_1, \ldots, a_{t-1}, z_t\}$, ACPOMDPs grant the critic privileged access to all information, including the underlying states. Thus, similar to CPOMDPs, ACPOMDPs are formulated by a tuple $(S, B, A, \tau, R_B, \mathbb{C}, d, \gamma)$, and aims to maximize the long-term belief expected reward $V_R(b)$ while bounding the long-term belief expected costs $V_{C_i}(b)$:

$$\pi_\star = \arg\max_{a \in A} V_R(b)$$
$$s.t. V_{C_i}(b) \leq b_i, \forall i \in [m] \tag{4}$$

Benefiting from the availability of the underlying states, at each time step, the critic receives and the action $a$ and the underlying state $s$, and updates the $V_R^*(s)$ using the following equation:

$$V_R^*(s) = \max_{a \in A} \left[ R(s,a) + \gamma \sum_{s'} P(s'|s,a) V_R^*(s') \right] \tag{5}$$

Consequently, the $V_R(b)$ and $V_{C_i}(b)$ in the optimization problem equation 4 are estimated using this updated $V_R^*(s)$:

$$V_R^*(b) = \sum_{s \in S} b(s) V_R^*(s) \tag{6}$$

Notice that the update of the $V_{C_i}(b)$ is not presented, which is equivalent to equation 6.

## 4.2 COMPARISON WITH CPOMDPs

We compare the different estimations of the $V_R(b)$ and $V_C(b)$ to demonstrate the superiority of ACPOMDPs. Since the $V_R(b)$ and $V_C(b)$ are equivalent in their estimations, we Collectively refer to them as $V(b)$. For clarity, we rewrite the $V(b)$ in CPOMDPs as $V_{sym}(b)$ and $V(b)$ in ACPOMDPs as $V_{asym}(b)$.

**Lemma 4.1** *Kaelbling et al. (1996) showed that the value function at time step $t$ can be expressed by a set of vectors: $\Gamma_t = \{\alpha_0, \alpha_1, \ldots, \alpha_m\}$. Each $\alpha$-vector represents an $|S|$-dimensional hyperplane, and defines the value function over a bounded region of the belief:*

$$V_t^*(b) = \max_{\alpha \in \Gamma_t} \sum_{s \in S} \alpha(s) b(s) \tag{7}$$

**Lemma 4.2** *Assume the state space $S$, action space $A$, and observation space $Z$ are finite. Let $|S|$, $|A|$, and $|Z|$ represent the number of states, actions, and observations, respectively. Let $|\Gamma_{t-1}|$ denote the size of the solution set for the value function $V_{t-1}(b)$ at time step $t-1$. The minimal number of elements required to express the value function $V_t(b)$ at time step $t$, denoted as $|\Gamma_t|$, grows as $|\Gamma_t| = O(|A||\Gamma_{t-1}|^{|Z|})$ (Pineau et al., 2006).*

We conclude that, at each time step $t$, the belief state space can be represented as a discrete representation space that exactly captures the value function $V_t(b)$. The size of this space is given by $|\Gamma_t| =$

$O(|A||\Gamma_{t-1}||Z|)$. Furthermore, as derived from equation 6, in the ACPOMDPs framework, the required size of the representation space is reduced to $|S|$. Clearly, $|S| \ll |\Gamma_t| = O(|A||\Gamma_{t-1}|^{|Z|})$.

Thus, ACPOMDPs significantly reduce the size of the representation space required to express the value function $V(b)$, eliminating observation-related uncertainties to the greatest extent possible. This, in turn, reduces the number of updates required for the critic to estimate the value function $V(b)$.

**Theorem 4.3** *Let $V_{asym}^*(b)$ and $V_{sym}^*(b)$ represent the optimal long-term expected values under the ACPOMDPs and CPOMDPs frameworks, respectively. Then, for all belief states $b \in B$, the inequality holds: $V_{asym}^*(b) \geq V_{sym}^*(b)$. (The proof is provided in Appendix A)*

The conclusion indicates that the ACPOMDPs framework, by leveraging asymmetric information, yields superior policies compared to the CPOMDPs framework. This is because, for any belief state $b \in B$, the optimal long-term expected reward under ACPOMDPs is always greater than or equal to that under CPOMDPs. Regarding safety, ACPOMDPs provide more accurate estimations due to additional information, while long-term expected costs under CPOMDPs are consistently lower or equal to those of ACPOMDPs. This implies that CPOMDPs tend to underestimate future safety risks.

## 5 METHODS

In this section, we introduce AsymDreamer, an algorithm grounded in the ACPOMDP framework that leverages privileged information to enhance both the agent's performance and safety. As Asym-Dreamer incorporates an observation world model, a privileged world model, and an actor-critic model, we emphasize the collaborative interaction between these components.

### 5.1 ASYMMETRIC WORLD MODEL LEARNING

The world models are parameterized with the learnable network parameter $\phi_o$ and $\phi_p$ respectively. At each time step $t$, the world models receive an observation $o_t$, an action $a_t$, and privileged information $p_t$ as inputs. Encoders map $o_t$ and $p_t$ to stochastic representations $z_t^o$ and $z_t^p$, respectively. The sequence models then use these representations, along with the action, to predict the next states $z_{t+1}^o$ and $z_{t+1}^g$, during which the recurrent states $h_t^o$ and $h_t^p$ are updated within the sequence models. We define the concatenation of $h_t$ and $z_t$ as the model state $s_t = \{h_t, z_t\}$. Finally, reward and cost decoders take the concatenation of $s_t^o$ and $s_t^p$ to predict rewards and costs, while observation and state decoders use $s_t^o$ and $s_t^p$ to predict the corresponding observations and states. In summary, the key model components are:

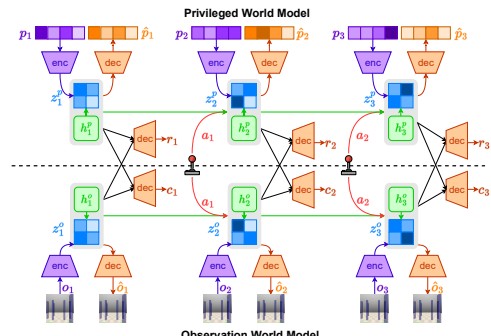

Figure 1: Asym World Model Learning

Observation World Model

$$
\begin{cases}
\text{Observation encoder: } z_t^o \sim E_{\phi_o}\left(z_t^o \mid h_t^o, o_t\right) \\
\text{Observation decoder: } \hat{o}_t \sim E_{\phi_o}\left(\hat{o}_t \mid s_t^o\right) \\
\text{Sequence model: } h_t^o, \hat{z}_t^o = E_{\phi_o}\left(z_t^o \mid s_{t-1}^o, a_{t-1}\right)
\end{cases}
$$

Privileged World Model

$$
\begin{cases}
\text{State encoder: } z_t^p \sim E_{\phi_p}\left(z_t^p \mid h_t^p, p_t\right) \\
\text{State decoder: } \hat{p}_t \sim E_{\phi_p}\left(\hat{p}_t \mid s_t^p\right) \\
\text{Reward decoder: } \hat{r}_t \sim E_{\phi_p}\left(\hat{r}_t \mid s_t^p, s_t^o\right) \\
\text{Cost decoder: } \hat{c}_t \sim E_{\phi_p}\left(\hat{c}_t \mid s_t^p, s_t^o\right) \\
\text{Sequence model: } h_t^p, \hat{z}_t^p = E_{\phi_p}\left(z_t^p \mid h_t^p, p_t\right)
\end{cases}
$$

**Trade-off Avoidance** As illustrated in Figure 1, the observation world model focuses exclusively on observation modeling, while the privileged world model emphasizes task-centric predictive capabilities. By separating observation modeling from task-centric prediction modeling, we can avoid the potential trade-off between these two tasks Ma et al. (2024). The specialization of the observation and privileged world models allows each component to excel in its respective domain without compromising the other. This synergistic approach ultimately results in improved overall performance.

**Information Sharing Mechanism** This asymmetric training structure provides the privileged world model with access to all the information from the observation world model. As shown in Figure 1, the privileged world model enhances its reward decoder and cost decoder by utilizing the union model state $s_t = \{s_t^o, s_t^p\}$. Experimental results indicate that while the privileged information contains all the information necessary for predicting costs and rewards, the incorporation of local observations continues to offer significant advantages.

**Information Maximization** This asymmetric training structure enables both the observation and privileged world models to capture the maximum amount of information. The observation world model, which focuses exclusively on observation modeling, is designed to capture more detailed observation information. Meanwhile, the privileged world model not only utilizes privileged information but also leverages the information from the observation world model. In addition, the privileged world model can converge more rapidly due to the low dimensionality of the privileged information, thereby accelerating the training of the critic model.

**Generalizability** Importantly, since most world models adhere to a common structural framework, the asymmetric training structure depicted in Figure 1 can be readily transferred to Bayesian world models Chua et al. (2018); Depeweg et al. (2018), latent variable world models Lee et al. (2020), RSSM-based world models Hafner et al. (2019), and Transformer-based world models Chen et al. (2022); van den Oord et al. (2018).

## 5.2 ASYMMETRIC ACTOR-CRITIC MODEL LEARNING

The actor and critic models learn purely from the imaginary rollouts predicted by world models. Specifically, at time step $t$, the actor model, parameterized with the learnable network parameter $\theta$, operates on the model state $s_t^o$ to predict the policy distribution $\pi_\theta(a_t \mid s_t^o)$. The critic models, on the other hand, operate on union model state $s_t$ to estimate the long-term expected returns $v_{\psi_r}(s_t)$ and $v_{\psi_r}(s_t)$. In summary, the key components of the actor-critic model are:

$$
\begin{aligned}
\text{Actor:} \quad & a_t \sim \pi_\theta(a_t \mid s_t^o) \\
\text{Reward Critic:} \quad & v_{\psi_r}(s_t) \approx \mathbb{E}_{\pi_\theta}\left[R_t^\lambda\right] \\
\text{Cost Critic:} \quad & v_{\psi_c}(s_t) \approx \mathbb{E}_{\pi_\theta}\left[C_t^\lambda\right]
\end{aligned}
\tag{8}
$$

**Synchronous Imagination** Due to the fact that the actor and the critic operate on two separate world models, a method must be developed to generate two imaginary rollouts that represent the same trajectory across both models. As shown in Figure 2, starting from representations of replayed inputs $s_1^o$ and $s_1^p$, for each time step $t$, the actor sample an action $a_t$ from the policy distribution $\pi_\theta(a_t \mid s_t^o)$ utilizing the $s_t^o$ form the observation world model, then each world model predict its next representations $s_{t+1}^o$ and $s_{t+1}^p$, along with predicted cost $\hat{c}_t$ and predicted reward $\hat{r}_t$, untill the time step $t$ reaches the imagination horizon $H = 15$. This synchronization of the imagination process across the two world models enables the actor and critic to learn from coherent simulated trajectories.

**Actor-Critic Model Learning** An imaginary trajectory $\{s_t^o, s_t^p, a_t, \hat{r}_t, \hat{c}_t\}_{1:H}$ is provided to the actor and the critics after the synchronous imagination process. Based on this trajectory, the critics can estimate the long-term expected returns $v_{\psi_r}(s_t)$ and $v_{\psi_r}(s_t)$ while the actor optimizes its policy according to a specified objective. Notably, there are no constraints on how long term expected returns are estimated and the optimization objective of the policy.

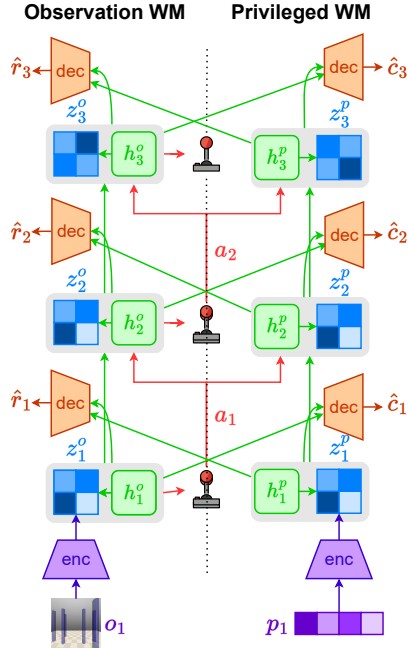

Figure 2: Synchronous Imagination

# 6 PRACTICAL IMPLEMENTATION

**World Model Implementation** The world model are implemented as a Recurrent State-Space Model (RSSM) Hafner et al. (2019), where the encoder and decoder that trained via variational auto-encoding Doersch (2021) method transforms the observation $o_t$ and privileged information $p_t$ into stochastic representations $z_t^o$, $z_t^p$, respectively. These stochastic representations $z_t$, together with action $a_t$ and recurrent state $h_t$ within the corresponding sequence model are used to predict next representation $z_{t+1}$, which are supervised by the dynamics loss. Meanwhile, the representations $z_{t+1}^o$ and $z_{t+1}^p$ are supervised by the regularization loss to ensure the representations $z_{t+1}^o$ and $z_{t+1}^p$ are predictable. The decoders are trained via the prediction loss. Specifically, the observation decoder is trained using Mean Squared Error (MSE) loss, while the reward and cost decoders are trained using the symlog loss.

$$\mathcal{L}(\phi_o)_{obs} = \sum_{t=1}^{T} \underbrace{\alpha_q^o \, \mathrm{KL}\left[z_t^o \parallel \mathrm{sg}(\hat{z}_t^o)\right]}_{\text{regularization loss}} + \underbrace{\alpha_p^o \, \mathrm{KL}\left[\mathrm{sg}(z_t^o) \parallel \hat{z}_t^o\right]}_{\text{dynamics loss}} - \underbrace{\beta_o^o \ln O_{\phi_o}\left(o_t \mid s_t^o\right)}_{\text{observation loss}} \tag{9}$$

$$\mathcal{L}(\phi_p)_{priv} = \sum_{t=1}^{T} \underbrace{\alpha_q^p \, \mathrm{KL}\left[z_t^p \parallel \mathrm{sg}(\hat{z}_t^p)\right]}_{\text{regularization loss}} + \underbrace{\alpha_p^p \, \mathrm{KL}\left[\mathrm{sg}(z_t^p) \parallel \hat{z}_t^p\right]}_{\text{dynamics loss}} - \underbrace{\beta_o^p \ln O_{\phi_p}\left(o_t \mid s_t^p\right)}_{\text{observation loss}}$$

$$- \underbrace{\beta_r^p \ln R_{\phi_p}\left(r_t \mid \mathrm{sg}(s_t^o) \parallel s_t^p\right)}_{\text{reward loss}} - \underbrace{\beta_c^p \ln C_{\phi_p}\left(c_t \mid \mathrm{sg}(s_t^o) \parallel s_t^p\right)}_{\text{cost loss}} \tag{10}$$

In the above expressions, $\mathrm{sg}(\cdot)$ represents the stop-gradient operator, and $\mathrm{KL}\left[\cdot\right]$ denotes the Kullback-Leibler (KL) divergence. Notably, It is worth noting that the reward and cost decoders use the union model state $s_t$ as an input to exploit observation and privileged information, and use $\mathrm{sg}(\cdot)$ on $s_t^o$ to prevent reward loss and cost loss from affecting the loss optimization of the observation world model.

**Actor-Critic Model Implementation** From given imaginary trajectory $\{s_t^o, s_t^p, a_t, \hat{r}_t, \hat{c}_t\}_{1:H}$ the bootstrapped $\mathrm{TD}(\lambda)$ value $R^\lambda(s_t)$ for the reward critic is calculated as follows:

$$R^\lambda(s_t) = \hat{r}_t + \gamma\left((1-\lambda)V_{\psi_r}(s_{t+1}) + \lambda R^\lambda(s_t)\right) \tag{11}$$

$$R^\lambda(s_T) = V_{\psi_r}(s_T) \tag{12}$$

These values are used to assess the long term expected reward, where $V_{\psi_r}(s_t)$ is approximated by the reward critic to consider the returns that beyond the imagination horizon $H$. Note that, we show here only the calculation of $R^\lambda(s_t)$, the calculation of $C^\lambda(s_t)$ is equivalent to equation 11. With the calculated $\mathrm{TD}(\lambda)$ values $R^\lambda(s_t)$ and $C^\lambda(s_t)$, we follow the equation 23 to define the policy optimization objective:

$$\mathcal{L}(\theta) = -\sum_{t=1}^{T} \mathrm{sg}\left(R^\lambda(s_t^o)\right) + \eta \mathrm{H}\left[\pi_\theta\left(a_t \mid s_t^o\right)\right] - \underbrace{\Psi\left(C^\lambda(s_t), \lambda_k^p, \mu_k\right)}_{\text{penalty term}} \tag{13}$$

This policy optimization objective encourages the actor to maximize the expected reward while simultaneously satisfying the specified safety constraints. The penalty term is formulated using the Augmented Lagrangian method Dai & Zhang (2021), which penalizes behaviors that violate safety constraints. Additionally, an entropy term is included in the objective to promote exploration. Further details regarding the policy optimization objective and the Augmented Lagrangian method can be found in Appendix D.

# 7 EXPERIMENTS

We conduct our experiments on Safety-Gymnasium, aiming to answer the following questions:

- Can the utilization of privileged information improve performance and safety?
- Are partial observations still necessary for critics when privileged information is available?
- How does our approach compare to existing approaches in terms of performance, sample efficiency and safety?

## 7.1 Safety-Gymnasium Benchmark

**SafetyQuadrotorGoal1** We find that all the tasks in Safety-Gymnasium are limited to a 2D plane, which hinders Safety-Gymnasium from evaluating a agent's ability to execute complex tasks in high-dimensional space. To fill this gap, we offer a new task, *SafetyQuadrotorGoal1*, to evaluate the model's capability to navigate in 3D space. As depicted in Figure 3, the blue cylinders represent hazards that the quadrotor must avoid, and the green sphere in the air denotes the navigation target for the quadrotor.

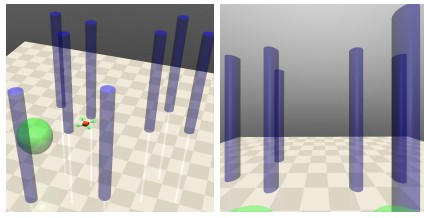

Figure 3: SafetyQuadrotorGoal1

The quadrotor has a four-dimensional action space, where each dimension corresponds to the force generated by each rotor. Further details are available in the Appendix B.

**Experimental Setup** In all our experiments, the agent observes a 64x64 pixel RGB image from the onboard camera. The task in our experiments is to navigate to the predetermined goal while avoiding collisions with other objects. The cost limit across all tasks is 2. We assess the task objective performance and safety using the following metrics proposed in:

- Average undiscounted episodic return for $E$ episode: $\hat{J}(\pi) = \frac{1}{E} \sum_{i=1}^{E} \sum_{t=0}^{T_{\text{ep}}} r_t$.

- Average undiscounted episodic cost return for $E$ episode: $\hat{J}_c(\pi) = \frac{1}{E} \sum_{i=1}^{E} \sum_{t=0}^{T_{\text{ep}}} c_t$.

We compute $\hat{J}(\pi)$ and $\hat{J}_c(\pi)$ by averaging the sum of costs and rewards across $E = 10$ evaluation episodes of length $T_{ep} = 1000$, without updating the agent's networks and discarding the interactions made during evaluation. The results for all methods are recorded once the agent reached $2M$ environment steps. Detailed designs of privileged information, descriptions of all baselines, and additional experiments can be found in Appendix E.

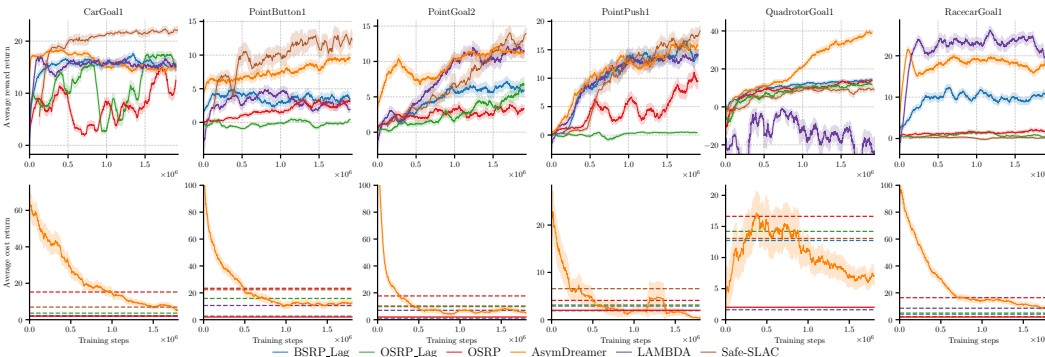

Figure 4: The experimental results for the Safety-Gymnasium Benchmark. The upper figures show the learning curves of **AsymDreamer** and the baseline algorithms. Meanwhile, the lower figures depict the learning curve of **AsymDreamer** and the final average cost return of the baseline algorithm marked with a dotted line for a clear comparison. The red solid line represents the cost limit for this task.

**Results** The findings of our experiments are summarized in Figure 4. As depicted in Figure 4, our algorithm demonstrates state-of-the-art performance across all tasks. **Safe-SLAC** is the sole algorithm that outperforms our approach regarding reward, it does so at the cost of incurring a high number of safety violations. Conversely, the **BSRP_Lag** algorithm is the sole algorithm that surpasses our approach in safety performance; however, it exhibits an excessively conservative behavior, resulting in consistently suboptimal task objective results. In contrast, our proposed algorithm consistently achieves very high rewards and excellent safety performance concurrently, enabling effective trade-offs between safety and task objective performance.

**Adaptability to Complex Scenarios** In particular, **AsymDreamer** significantly outperforms alternative algorithms in both task performance and safety on the *SafetyQuadrotorGoal1* task. This task, which requires navigating a 3D space, presents a larger state space and increased partial observability due to the agent needing to exert more effort in observing its environment. However, by

leveraging privileged information, our approach minimizes partial observability, giving the agent a significant advantage in terms of available information. This superior performance is consistent with our conclusion in Section 4.2, where we highlighted the agent's ability to learn more effective policies by leveraging privileged information.

## 7.2 ABLATION STUDY

Our ablation study includes the following settings: (1) **AsymDreamer:** The full version of AsymDreamer, where the critic leverages the model states from both the Observation World Model and Privileged World Model. (2) **AsymDreamer(S):** In this variant, the critic takes only the model state of the Privileged World Model as input. (3) **AsymDreamer(O):** Here, the critic takes solely the model state of the Observation World Model as input. This variant corresponds to the **BSRP_Lag** in SafeDreamer. (4) **DreamerV3:** The default DreamerV3 setup, where we remove the last term of equation 13 on the basis of **AsymDreamer(O)**.

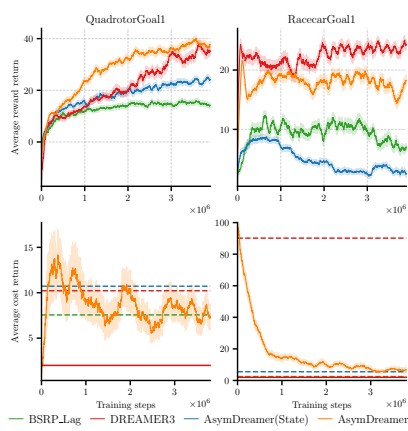

Figure 5: Results in ablation study

**Partial Observations Remain Valuable.** As depicted in Figure 5, **AsymDreamer(S)**, which utilizes privileged information, does not demonstrate a significant improvement in task objectives compared to **AsymDreamer(O)** and may even perform worse. This is primarily because relying solely on privileged information for value estimation causes the model to overlook the information gained from observing the environment. Consequently, this results in a one-sided pursuit of reward maximization, ultimately leading to lower performance. Additionally, **AsymDreamer(S)** exhibits inadequate safety in the *SafetyQuadrotorGoal1* task. We identify two reasons for this: (1) The cost distribution in this task is highly unbalanced. (2) The cost decoder, which relies on privileged information as input, must learn additional information to effectively estimate the cost function.

**Privileged Information Leads to Significant Improvements.** As depicted in Figure 5, **AsymDreamer**, which leverages both partial observations and privileged information, achieves significantly superior performance compared to the other settings. This suggests that partial observations, even in the presence of comprehensive privileged information, continue to provide valuable complementary information that enhances the overall system capabilities. Finally, we compare our **AsymDreamer**, which incorporates safety constraints, with **DreamerV3**, which neglects such constraints. Remarkably, on the *SafetyQuadrotorGoal1* task, our model outperforms **DreamerV3** in terms of task objective performance while simultaneously achieving the lowest safety violation. To the best of our knowledge, we are the first method to achieve this feat.

## 8 CONCLUSION

We present **AsymDreamer**, a model-based reinforcement learning approach specifically designed for partially observable environments with safety constraints. AsymDreamer employs an asymmetric architecture, where the actor constructs a world model based on the agent's partial observations, while the critic leverages a privileged world model that incorporates additional privileged information. This approach is formalized within the *Asymmetric Constrained Partially Observable Markov Decision Processes* (ACPOMDP) framework, offering theoretical advantages in addressing the challenges of partial observability and safety. To ensure compliance with safety constraints, AsymDreamer integrates the Lagrangian method to handle constrained optimization problems. AsymDreamer demonstrates competitive performance across benchmarks and exhibits strong adaptability in complex scenarios. However, we have identified certain limitations in the current design. Specifically, some forms of privileged information do not significantly enhance the performance of the cost predictor, limiting their overall contribution to the model. Given the critical importance of world models, future work could explore techniques to train more robust models capable of effectively capturing sparse signals.

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

## A  PROOF

In this section, we prove Theorem 4.3. First, we donate the optimal long-term values in the belief space under the ACPOMDPs framework as $V^*_{asym}(b)$:

$$V^*_{asym}(b) = \sum_{s \in S} b(s) V^*(s)$$

$$V^*(s) = \max_{a \in A} \left[ R(s,a) + \gamma \sum_{s'} P(s'|s,a) V^*(s') \right] \tag{14}$$

where $V^*(s)$ represents the optimal long-term values in the state space. Similar to $V^*_{asym}(b)$, the optimal long-term values in the belief space under the CPOMDPs framework are represented as $V^*_{sym}(b)$:

$$V^*_{sym}(b) = \max_{a \in A} \left[ R(b,a) + \gamma \sum_{z \in Z} Pr(z|b,a) V^*_{sym}(b^{a,z}) \right] \tag{15}$$

Since

$$Pr(z|b,a) = \sum_{s' \in S} Pr(z|a,s') \sum_{s \in S} Pr(s'|s,a) b(s) \tag{16}$$

$V^*_{sym}(b)$ can be rewritted as:

$$V^*_{sym}(b) = \max_{a \in A} \left[ R(b,a) + \gamma \sum_{z \in Z} V^*_{sym}(b^{a,z}) \sum_{s' \in S} Pr(z|a,s') \sum_{s \in S} P(s'|s,a) b(s) \right]$$

$$= \max_{a \in A} \left[ R(b,a) + \gamma \sum_{z \in Z} V^*_{sym}(b^{a,z}) \sum_{s \in S} \sum_{s' \in S} Pr(z|a,s') P(s'|s,a) b(s) \right] \tag{17}$$

*Proof.*  According to Lemma 4.1, $V^*(s')$ can be expressed by a set of vectors: $\Gamma_t = \{\alpha_0, \alpha_1, \dots, \alpha_m\}$. As a result, $V^*(s')$ can be rewrite as the following equation:

$$V^*(s) = \max_{a \in A} \left[ R(s,a) + \gamma \sum_{s'} P(s'|s,a) \max_{\alpha \in \Gamma_t} \alpha(s') \right] \tag{18}$$

Similarly, $V^*_{sym}(b)$ can be rewritten as:

$$V^*_{sym}(b) = \max_{a \in A} \left[ \sum_{s \in S} R(s,a) b(s) + \gamma \sum_{z \in Z} \max_{\alpha \in \Gamma_{t-1}} \sum_{s \in S} \sum_{s' \in S} Pr(z|a,s') P(s'|s,a) b(s) \alpha(s') \right] \tag{19}$$

Then:

$$V^*_{asym}(b) = \sum_{s \in S} b(s) V^*(s)$$

$$= \sum_{s \in S} b(s) \max_{a \in A} \left[ R(s,a) + \gamma \sum_{s'} P(s'|s,a) \max_{\alpha \in \Gamma_t} \alpha(s') \right]$$

$$\geq \max_{a \in A} \left[ \sum_{s \in S} b(s) R(s,a) + \gamma \sum_{s \in S} b(s) \sum_{s' \in S} P(s'|s,a) \max_{\alpha \in \Gamma_t} \alpha(s') \right] \tag{20}$$

$$= \max_{a \in A} \left[ R(b,a) + \gamma \sum_{s \in S} \sum_{s' \in S} \sum_{z \in Z} Pr(z|a,s') P(s'|s,a) b(s) \max_{\alpha \in \Gamma_t} \alpha(s') \right]$$

$$\geq \max_{a \in A} \left[ R(b,a) + \gamma \sum_{z \in Z} \max_{\alpha \in \Gamma_t} \sum_{s \in S} \sum_{s' \in S} Pr(z|a,s') P(s'|s,a) b(s) \alpha(s') \right]$$

$$= V^*_{sym}(b)$$

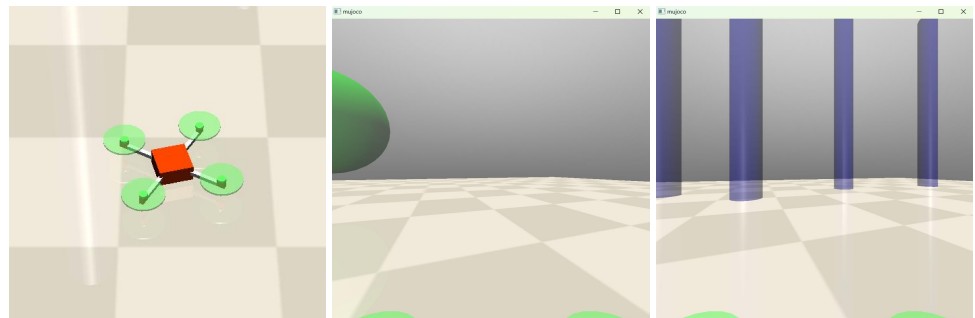

Figure 6: Quadrotor

# B SAFETYQUADROTORGOAL1

In this section, we give detailed design of the *SafetyQuadrotorGoal1* task.

## B.1 SCENE GENERATION

**Hazardous Areas (Hazard).** As shown in Figure 4, the hazardous area is presented as a cylinder with a radius of 0.1 and a height of is 1.0.

**Goal.** the goal is presented as a sphere with a radius of 0.3.

**Generation Algorithm.** A random generation algorithm is used to generate the target based on the $(x, y)$ axis coordinates of the hazardous area, the target's Location. As shown in Figure 4, both the hazardous area and the target are set with a keepout value of 0.4. Each object generates a position whenever the distance between the generated position and the generated object is less than or equal to the keepout value. Whenever the distance between the generated position and the generated object is less than or equal to the keepout value, the position is regenerated. After all object positions are generated, the target is randomly generated with $z$-axis coordinates between 0.3 and 1.7.

## B.2 REWARD FUNCTION

Using dense rewards to guide learning and encode tasks to reach a goal through obstacle avoidance. At each time step, the reward is computed as:

$$r_t = r_t^{alive} + r_t^{prog} + r_t^{perc} + r_t^{goal} - r_t^{cmd} - r_t^{a\text{angular}} \tag{21}$$

$$r_t^{\text{alive}} = \lambda_1 (\mathrm{d}_{t-1} - \mathrm{d}_t)$$
$$r_t^{\text{perc}} = \lambda_2 \exp\left(-\delta_{\text{cam}}^4\right)$$
$$r_t^{\text{cmd}} = \lambda_3 ||a_t|| + \lambda_4 ||a_t - a_{t-1}||^2$$
$$r_t^{\text{alive}} = \begin{cases} 0.01 & \text{if alive} \\ 0 & \text{otherwise} \end{cases}$$
$$r_t^{\text{angular}} = -\lambda_5 ||\omega||$$
$$r_t^{\text{goal}} = \begin{cases} 4.0 & \text{if goal} \\ 0 & \text{otherwise} \end{cases}$$

where $\delta_{\text{cam}}$ is the angle between the optical axis of the camera and the vector pointing from the UAV to the target. The hyperparameters $\lambda_1 = 0.5$, $\lambda_2 = 0.025$, $\lambda_3 = 0.0005$, and $\lambda_4 = 0.0002$ are chosen empirically and weighed against the speed and smoothness of the strategy.

The reward $r_{\text{alive}}$ encourages the survival of the UAV and prevents it from crashing to the ground or becoming unable to take off. The progress bonus $r_{\text{prog}}$ encourages fast flight to maximize the number of successful flights.

## C HYPERPARAMETERS

### C.1 ASYMDREAMER AND SAFEDREAMER

AsymDreamer is implemented based on SafeDreamer, so they follow the same setting.

Table 1: Hyperparameters

| Name | Symbol | Value |
|------|--------|-------|
| **World Model** | | |
| Number of latent classes | | 48 |
| Classes per latent | | 48 |
| Batch size | $B$ | 64 |
| Batch length | $T$ | 16 |
| Learning rate | | $10^{-4}$ |
| Coefficient of KL divergence in loss | $\alpha_q, \alpha_p$ | 0.1, 0.5 |
| Coefficient of decoder in loss | $\beta_o, \beta_r, \beta_c$ | 1.0, 1.0, 1.0 |
| **Planner** | | |
| Planning horizon | $H$ | 15 |
| Number of samples | $N_{\pi N}$ | 500 |
| Mixture coefficient | $M$ | 0.05 |
| $N_{\pi\theta} = M \cdot N_{\pi N}$ | | |
| Number of iterations | $J$ | 6 |
| Initial variance | $\sigma_0$ | 1.0 |
| **PID Lagrangian** | | |
| Proportional coefficient | $K_p$ | 0.01 |
| Integral coefficient | $K_i$ | 0.1 |
| Differential coefficient | $K_d$ | 0.01 |
| Initial Lagrangian multiplier | $\lambda_{p0}$ | 0.0 |
| Lagrangian upper bound | | 0.75 |
| Maximum of $\lambda_p$ | | |
| **Augmented Lagrangian** | | |
| Penalty term | $\nu$ | $5^{-9}$ |
| Initial penalty multiplier | $\mu_0$ | $1^{-6}$ |
| Initial Lagrangian multiplier | $\lambda_{p0}$ | 0.01 |
| **Actor Critic** | | |
| Sequence generation horizon | | 15 |
| Discount horizon | $\gamma$ | 0.997 |
| Reward lambda | $\lambda_r$ | 0.95 |
| Cost lambda | $\lambda_c$ | 0.95 |
| Learning rate | | $3 \cdot 10^{-5}$ |
| **General** | | |
| Number of other MLP layers | | 5 |
| Number of other MLP layer units | | 512 |
| Train ratio | | 512 |
| Action repeat | | 4 |

### C.2 SAFE-SLAC

Hyperparameters for Safe-SLAC. We maintain the original hyperparameters unchanged, with the exception of the action repeat, which we adjust from its initial value of 2 to 4.

Table 2: Hyperparameters for Safe-SLAC

| Name | Value |
|---|---|
| Length of sequences sampled from replay buffer | 15 |
| Discount factor | 0.99 |
| Cost discount factor | 0.995 |
| Replay buffer size | $2 \times 10^5$ |
| Latent model update batch size | 32 |
| Actor-critic update batch size | 64 |
| Latent model learning rate | $1 \times 10^{-4}$ |
| Actor-critic learning rate | $2 \times 10^{-4}$ |
| Safety Lagrange multiplier learning rate | $2 \times 10^{-4}$ |
| Action repeat | 4 |
| Cost limit | 2.0 |
| Initial value for $\alpha$ | $4 \times 10^{-3}$ |
| Initial value for $\lambda$ | $2 \times 10^{-2}$ |
| Warmup environment steps | $60 \times 10^3$ |
| Warmup latent model training steps | $30 \times 10^3$ |
| Gradient clipping max norm | 40 |
| Target network update exponential factor | $5 \times 10^{-3}$ |

## C.3 LAMBDA

Hyperparameters for LAMBDA. We maintain the original hyperparameters unchanged, with the exception of the action repeat, which we adjust from its initial value of 2 to 4.

Table 3: Hyperparameters for LAMBDA

| Name | Value |
|---|---|
| Sequence generation horizon | 15 |
| Sequence length | 50 |
| Learning rate | $1 \times 10^{-4}$ |
| Burn-in steps | 500 |
| Period steps | 200 |
| Models | 20 |
| Decay | 0.8 |
| Cyclic LR factor | 5.0 |
| Posterior samples | 5 |
| Safety critic learning rate | $2 \times 10^{-4}$ |
| Initial penalty | $5 \times 10^{-9}$ |
| Initial Lagrangian | $1 \times 10^{-6}$ |
| Penalty power factor | $1 \times 10^{-5}$ |
| Safety discount factor | 0.995 |
| Update steps | 100 |
| Critic learning rate | $8 \times 10^{-5}$ |
| Policy learning rate | $8 \times 10^{-5}$ |
| Action repeat | 4 |
| Discount factor | 0.99 |
| TD($\lambda$) factor | 0.95 |
| Cost limit | 2.0 |
| Batch size | 32 |

# D   THE AUGMENTED LAGRANGIAN

The Augmented Lagrangian method incorporates the safety constraints into the optimization process by adding a penalty term to the objective function. This allows the actor model to optimize the expected reward while simultaneously satisfying the specified safety constraints. As a result, by adopting the Augmented Lagrangian method, we transform the optimization problem in equation 4 into an unconstrained optimization problem:

$$\max_{\pi \in \Pi} \min_{\boldsymbol{\lambda} \geq 0} \left[ R(\pi) - \sum_{i=1}^{C} \lambda^i \left( C_i(\pi) - b_i \right) + \frac{1}{\mu_k} \sum_{i=1}^{C} \left( \lambda^i - \lambda_k^i \right)^2 \right] \tag{22}$$

where $\lambda^i$ are the Lagrange multipliers, each corresponding to a safety constraint measured by $C_i(\pi)$, and $\mu_k$ is a non-decreasing penalty term corresponding to gradient step $k$. We take gradient steps of the following unconstrained objective:

$$\tilde{R}(\pi; \boldsymbol{\lambda}_k, \mu_k) = R(\pi) - \sum_{i=1}^{C} \Psi(C_i(\pi), \lambda_k^i, \mu_k) \tag{23}$$

We define $\Delta_i = C_i(\pi) - b_i$. The update rules for the penalty term $\Psi(C_i(\pi), \lambda_k^i, \mu_k)$ and the Lagrange multipliers $\lambda^i$ are as follow:

$$\forall i \in [m] : \Psi(C_i(\pi), \lambda_k^i, \mu_k), \lambda_{k+1}^i = \begin{cases} \lambda_k^i \Delta_i + \frac{\mu_k}{2} \Delta_i^2, \lambda_k^i + \mu_k \Delta_i & \text{if } \lambda_k^i + \mu_k \Delta_i \geq 0 \\ -\frac{(\lambda_k^i)^2}{2\mu_k}, 0 & \text{otherwise.} \end{cases} \tag{24}$$

# E   EXPERIMENTS

## E.1   PRIVILEGED INFORMATION DESIGN

In different tasks, it is necessary to customise the use of different privileged information, and different privileged information will have different impacts, we show our privileged information settings in our experiments.

| Privileged Information Name | Dimension | Description |
|---|---|---|
| hazards | $(n, 2)$ | Represents the relative positions of hazards in the environment, containing 2D coordinates $[x, y]$. |
| velocimeter | $(2, )$ | Provides the agent's velocity information in three-dimensional space $[v_x, v_y]$. |
| accelerometer | $(1, )$ | Provides the agent's acceleration information in three-dimensional space $[a_x]$. |
| gyro | $(1, )$ | Provides the agent's angular velocity information $[\omega_z]$. |
| goal | $(2, )$ | Represents the relative coordinates of the target position that the agent needs to reach $[x_{goal}, y_{goal}]$. |
| robot_m | $(2, )$ | Represents the rotation matrix of the robot, describing the robot's orientation and rotation in three-dimensional space. |
| push_box | $(n, 2)$ | Represents the relative positions of push_box in the environment, containing 2D coordinates $[x, y]$. |
| push_box_mat | $(2, )$ | Represents the rotation matrix of the push_box, describing the robot's orientation and rotation in three-dimensional space. |
| push_box_vel | $(2, )$ | Provides the push_box's velocity information in three-dimensional space $[v_x, v_y]$. |

Table 4: Privileged Information: SafetyPointPush1

| Privileged Information Name | Dimension | Description |
|---|---|---|
| hazards | $(n, 3)$ | Represents the relative positions of hazards in the environment, containing 3D coordinates $[x, y, z]$. |
| velocimeter | $(3, )$ | Provides the agent's velocity information in three-dimensional space $[v_x, v_y, v_z]$. |
| accelerometer | $(3, )$ | Provides the agent's acceleration information in three-dimensional space $[a_x, a_y, a_z]$. |
| gyro | $(3, )$ | Provides the agent's angular velocity information $[\omega_x, \omega_y, \omega_z]$. |
| goal | $(3, )$ | Represents the relative coordinates of the target position that the agent needs to reach $[x_{goal}, y_{goal}, z_{goal}]$. |
| robot_m | $(3, 3)$ | Represents the rotation matrix of the robot, describing the robot's orientation and rotation in three-dimensional space. |
| past_1_action | $(4, )$ | Represents the action information from the previous time step. |
| past_2_action | $(4, )$ | Represents the action information from the second-to-last time step. |
| past_3_action | $(4, )$ | Represents the action information from the third-to-last time step. |
| euler | $(2, )$ | Represents the agent's pose information given in Euler angles $[roll, pitch]$. |

Table 5: Privileged Information: SafetyQuadrotorGoal1

| Privileged Information Name | Dimension | Description |
| --- | --- | --- |
| hazards | $(n, 2)$ | Represents the relative positions of hazards in the environment, containing 2D coordinates $[x, y]$. |
| vases | $(n, 2)$ | Represents the relative positions of vases in the environment, containing 2D coordinates $[x, y]$. |
| velocimeter | $(2, )$ | Provides the agent's velocity information in three-dimensional space $[v_x, v_y, v_z]$. |
| accelerometer | $(1, )$ | Provides the agent's acceleration information in three-dimensional space $[a_x]$. |
| gyro | $(1, )$ | Provides the agent's angular velocity information $[\omega_z]$. |
| goal | $(2, )$ | Represents the relative coordinates of the target position that the agent needs to reach $[x_{\text{goal}}, y_{\text{goal}}]$. |

Table 6: Privileged Information: SafetyPointGoal2

| Privileged Information Name | Dimension | Description |
| --- | --- | --- |
| hazards | $(n, 2)$ | Represents the relative positions of hazards in the environment, containing 2D coordinates $[x, y]$. |
| vases | $(n, 2)$ | Represents the relative positions of vases in the environment, containing 2D coordinates $[x, y]$. |
| velocimeter | $(2, )$ | Provides the agent's velocity information in three-dimensional space $[v_x, v_y, v_z]$. |
| accelerometer | $(1, )$ | Provides the agent's acceleration information in three-dimensional space $[a_x]$. |
| gyro | $(1, )$ | Provides the agent's angular velocity information $[\omega_z]$. |
| goal | $(2, )$ | Represents the relative coordinates of the target position that the agent needs to reach $[x_{\text{goal}}, y_{\text{goal}}]$. |

Table 7: Privileged Information: SafetyRacecarGoal1

| Privileged Information Name | Dimension | Description |
| --- | --- | --- |
| hazards | $(n, 2)$ | Represents the relative positions of hazards in the environment, containing 2D coordinates $[x, y]$. |
| vases | $(n, 2)$ | Represents the relative positions of vases in the environment, containing 2D coordinates $[x, y]$. |
| velocimeter | $(2, )$ | Provides the agent's velocity information in three-dimensional space $[v_x, v_y, v_z]$. |
| accelerometer | $(1, )$ | Provides the agent's acceleration information in three-dimensional space $[a_x]$. |
| gyro | $(1, )$ | Provides the agent's angular velocity information $[\omega_z]$. |
| goal | $(2, )$ | Represents the relative coordinates of the target position that the agent needs to reach $[x_{\text{goal}}, y_{\text{goal}}]$. |

Table 8: Privileged Information: SafetyCarGoal1

| Privileged Information Name | Dimension | Description |
| --- | --- | --- |
| hazards | $(n, 2)$ | Represents the relative positions of hazards in the environment, containing 2D coordinates $[x, y]$. |
| velocimeter | $(2, )$ | Provides the agent's velocity information in three-dimensional space $[v_x, v_y]$. |
| accelerometer | $(1, )$ | Provides the agent's acceleration information in three-dimensional space $[a_x]$. |
| gyro | $(1, )$ | Provides the agent's angular velocity information $[\omega_z]$. |
| goal | $(2, )$ | Represents the relative coordinates of the target position that the agent needs to reach $[x_{\text{goal}}, y_{\text{goal}}]$. |
| gremlins | $(n, 2)$ | Represents the relative positions of gremlins in the environment, containing 2D coordinates $[x, y]$. |
| buttons | $(n, 2)$ | Represents the relative positions of buttons in the environment, containing 2D coordinates $[x, y]$. |

Table 9: Privileged Information: SafetyPointButton1

## E.2 EXPERIMENTAL SETUP

**Setup.** Our experiments were conducted using the following configuration: a single A100-PCIE-40GB GPU (40GB), a 10 vCPU Intel Xeon Processor (Skylake, IBRS), and 72GB of memory.

**Baselines.** We compared AsymDreamer to several competitive baselines to demonstrate the superior results of using privileged information. The baselines include: 1. **Dreamerv3:** (Hafner et al., 2023) A general algorithm that could master diverse domains with fixed hyperparameters. 2. **BSRP_Lag:** (Huang et al., 2024) Integrates Dreamerv3 with the Lagrangian methods. 3. **OSRP:** Integrates Dreamerv3 with the CCEM methods. 4. **OSRP_Lag:** Integrates Dreamerv3 and the Lagrangian methods with the CCEM methods. 5. **LAMBDA:** (As et al., 2022) A novel model-based approach utilizes Bayesian world models and the Lagrangian methods. 6. **Safe-SLAC:** (Hogewind et al., 2022) Integrates SLAC with the Lagrangian methods. 7. **CPO:** Achiam et al. (2017) the first general-purpose policy search algorithm for constrained reinforcement learning with guarantees for near-constraint satisfaction at each iteration. 8. **PPO_Lag:** Achiam & Amodei (2019) Integrates PPO with the Lagrangian methods. 9. **TRPO_Lag:** Integrates TRPO with the Lagrangian methods. 10. **FOCOPS:** Zhang et al. (2020) initially determines the optimal update policy by addressing a constrained optimization problem within the nonparameterized policy space. Subsequently, FOCOPS projects the update policy back into the parametric policy space. Notably, the OSRP, OSRP_Lag and BSRP_Lag are three algorithms proposed by SafeDreamer.

## E.3 MODEL-FREE

We also compare our results with several model-free algorithms. In our experiments, the baseline algorithm results are derived from SafePO Ji et al. (2023a), which is configured with a cost limit of 25, whereas AsymDreamer is set with a cost limit of 2. The comparison results are presented in the Table 10.

| | CPO | | FOCOPS | | PPO_Lag | | TRPO_Lag | | AsymDreamer(Ours) | |
|---|---|---|---|---|---|---|---|---|---|---|
| Tasks | Reward | Cost | Reward | Cost | Reward | Cost | Reward | Cost | Reward | Cost |
| CarGoal1 | **23.2±1.9** | 28.2±4.6 | 21.5±0.0 | 28.1±0.0 | 13.8±3.3 | 23.4±10.8 | 22.2±3.9 | 26.2±6.1 | 14.5±0.5 | **4.2±1.2** |
| PointButton1 | 6.8±1.6 | 29.8±6.1 | 8.9±10.7 | **10.2±4.5** | 4.0±1.4 | 28.2±13.8 | 7.5±1.4 | 26.3±6.0 | **9.6±3.5** | 12.5±2.3 |
| PointPush1 | 4.8±0.0 | 25.5±0.0 | 0.7±0.7 | 23.0±21.1 | 0.6±0.3 | 26.2±25.1 | 0.6±0.1 | 21.7±11.2 | **15.6±0.8** | **0.4±0.2** |
| RacecarGoal1 | 10.4±1.2 | 29.4±7.0 | 4.5±2.2 | 93.7±33.3 | 2.3±2.1 | 28.3±12.7 | 9.5±3.0 | 25.1±5.7 | **18.2±1.2** | 6.8±1.2 |
| Average | 11.3 | 28.3 | 8.9 | 38.8 | 5.2 | 26.6 | 9.9 | 24.9 | **14.5** | **6.0** |

Table 10: Comparison with model-free algorithms

As illustrated in Table 10, the model-free algorithm encounters difficulties in achieving higher rewards, even with a more relaxed cost threshold. This challenge arises partly from the inability of these algorithms to leverage historical information, as well as their lack of access to essential privileged information. In contrast, AsymDreamer significantly outperforms the baseline algorithm in both reward and safety, owing to its effective utilization of privileged information.

## E.4 TRAINING EFFICIENCY

Since our model requires modeling two world models, we will compare the training efficiencies of the different algorithms to determine if the addition of a world model results in a significant increase in training time.

| | LAMBDA | Safe-SLAC | OSRP | OSRP_Lag | BSRP_Lag | **Ours** | **Incremental** |
|---|---|---|---|---|---|---|---|
| CarGoal1 | 34.1 | 18.3 | 22.5 | 18.9 | 28.2 | 37.8 | 34.0% |
| PointButton1 | 28.4 | 18.5 | 19.5 | 31.4 | 15.5 | 26.4 | 70.3% |
| PointGoal2 | 31.1 | 18.1 | 24.9 | 22.5 | 15.4 | 19.4 | 25.9% |
| PointPush1 | 36.7 | 16.1 | 20.1 | 18.3 | 13.4 | 21.9 | 63.4% |
| QuadrotorGoal1 | 34.6 | 15.9 | 33.1 | 42.1 | 21.6 | 36.6 | 69.4% |
| RacecarGoal1 | 34.1 | 19.5 | 20.5 | 20.7 | 28.0 | 44.3 | 58.2% |
| Average | 33.2 | 17.7 | 23.4 | 25.7 | 20.4 | 31.1 | 52.4% |

Table 11: Comparison of Training Time

Table 11 displays the hours required to train all algorithms for 2 million iterations on each task. The comparison between BSRP_Lag and AsymDreamer reveals that the inclusion of an additional world model in AsymDreamer resulted in an average increase of 50% in training hours.

