# OpenReview forum: "AsymDreamer: Safe Reinforcement Learning From Pixels with Privileged World Models"
_ICLR.cc/2025/Conference — Submitted to ICLR 2025_

### Official Review · Reviewer_xknH · 2024-10-28

**Soundness:** 3
**Presentation:** 3
**Contribution:** 2
**Rating:** 5
**Confidence:** 3

**Summary:**

In this work, the authors address the challenge of exploiting asymmetric inputs under the CPOMDPs framework. They propose AsymDreamer, a new algorithm that uses privileged information to build a world model for the critic. They also introduce the ACPOMDPs framework, an extension of CPOMDPs allowing asymmetric inputs for the actor and critic. Theoretically, asymmetric inputs reduce critic updates and lead to a better policy. AsymDreamer constructs two world models, one for the actor based on historical information and another for the critic with privileged information. It is integrated with the Lagrangian method and shows competitive performance on the Safety-Gymnasium benchmark and strong adaptability to complex scenarios.

**Strengths:**

* The proposed method of combining DreamerV3 with two world models and Lagrangian methods, which is novel and well-motivated.
* The paper presents empirical results that compare AsymDreamer with many relevant model-based and model-free baselines in several tasks, achieving competitive performance on the Safety-Gymnasium benchmark.
* The paper is mostly well-written.

**Weaknesses:**

* The work is built on top of SafeDreamer, incrementally adding another world model for critic, and differences in terms of the loss optimized are not sufficiently highlighted.
* The DreamerV3 is basically combinations of existing methods like CPOMDPs, Augmented Lagrangian, and RSSM. While this is not a problem in itself as they are common in model-based reinforcement learning, there is very little discussion on why these particular methods were chosen and what possible alternatives from the literature exist and whether they might yield better results.
* Lack of introduction to baseline algorithms. And in the PointGoal2 scenario, the performance results seem to be different from SafeDreamer reported in the paper.
* Stability verification needs to be considered. How the hyperparameters in equation (8), (9) are selected, and certain ablation experiments need to be conducted.
* Whether modeling two world models will lead to a significant increase in learning time and the effectiveness of model learning also need to be considered.
* Privileged information and partial observations should be considered in more environments. A single scenario in QuadrotorGoal1 cannot be trusted to determine whether privileged information enhances performance.

**Questions:**

It would be great if the authors could address the weaknesses I outlined above.

---

> ### Author Response · Authors · 2024-11-19
> **Response to Reviewer xknH (1)**
>
> Thank you for your thorough review and valuable suggestions regarding this paper. Your suggestions have greatly helped us to improve our paper. **The modifications can be seen in the new version of the PDF.** In response to your concerns, we have provided detailed replies, added more analysis, and included more ablation studies. We hope these address your concerns and would appreciate it if you could adjust your scores accordingly!
>
> ## Weakness 1
> **The work is built on top of SafeDreamer, incrementally adding another world model for critic, and differences in terms of the loss optimized are not sufficiently highlighted.**
>
> **Re:**
> There are two primary differences between SafeDreamer and our work regarding loss optimization:
> 1. The observation world model does not include reward and cost decoders, and consequently lacks corresponding prediction losses.
> 2. In the privileged world model, the reward and cost decoders utilize the model state from the world model as input. However, a stop gradient approach is employed, which prevents the gradient from the privileged world model from being transmitted to the observation world model. This ensures that the observation world model remains focused solely on observation modeling.
>
> At the same time, we want to emphasize that our key contribution is not adding a world model to SafeDreamer. Our key contribution is the proposal of an asymmetric training structure that effectively exploits privileged information in model-based reinforcement learning. This approach offers four distinct benefits:
> 1. It enables us to avoid potential trade-offs between observational modeling and task-centric predictive modeling.
> 2. The asymmetric training structure allows the privileged world model to access all information from the observational world model, thereby enhancing its reward decoder and cost decoder.
> 3. This structure facilitates both the observed world model and the privileged world model in capturing the maximum amount of information.
> 4. The asymmetric training structure can be seamlessly transferred to implementations of other world models.
>
> The theoretical analysis of ACPOMDP is also an important part of our work. Our comparative analysis with CPOMDP shows that significant improvements can be achieved by using an asymmetric structure:
> 1. It reduces the number of updates required by the critic to estimate the value function.
> 2. It leads to more optimal policies compared to the standard CPOMDPs framework.
>
> ## Weakness 2
> **The DreamerV3 is basically combinations of existing methods like CPOMDPs, Augmented Lagrangian, and RSSM. While this is not a problem in itself as they are common in model-based reinforcement learning, there is very little discussion on why these particular methods were chosen and what possible alternatives from the literature exist and whether they might yield better results.**
>
> **Re:**
> Thank you very much for your question, and we will provide further clarification of your concerns. We implemented our method using RSSM and Augmented Lagrangian methods primarily because both approaches are highly popular and powerful, widely accepted in the safety reinforcement learning community, and supported by a substantial body of research.
>
> **However, we would like to emphasize that our approach is not dependent on these specific techniques.** In fact, our method can be easily adapted to Bayesian world models, Transformer-based world models, and latent variable world models. Furthermore, the augmented Lagrangian method can be entirely replaced with alternative methods, such as LBSGD.
>
> Therefore, our primary focus is on whether the inclusion of privileged information enhances the algorithm's effectiveness, rather than solely on the performance improvements offered by the other components.
>
> ## Weakness 3
> **Lack of introduction to baseline algorithms. And in the PointGoal2 scenario, the performance results seem to be different from SafeDreamer reported in the paper.**
>
> **Re:**
> Thank you very much for your suggestion. **We have added a description of the baseline in Appendix F.**
>
> For the SafeDreamer experiments, we did not intend to degrade the performance of any baseline. We executed the unmodified version of the open-source repository from [SafeDreamer GitHub](https://github.com/PKU-Alignment/SafeDreamer) to ensure a fair comparison. We also reviewed relevant literature to support our experiments. Reference [1] employed the SafeDreamer algorithm and achieved performance comparable to ours after 2 million runs in the *SafetyPointGoal2* scenario, with a final reward of approximately 7.
>
> **We've asked the authors of SafeDreamer and they use RGB images from both the front and rear cameras as input, but we only used one,  which should account for the difference in results.**
>
> [1] Cao, C., Xin, Y., Wu, S., He, L., Yan, Z., Tan, J., and Wang, X. (2024). Fosp: Fine-tuning offline safe policy through world models.

---

> > ### Author Response · Authors · 2024-11-19
> > **Response to Reviewer xknH (2)**
> >
> > ## Weakness 4
> > **Stability verification needs to be considered. How the hyperparameters in equation (8), (9) are selected, and certain ablation experiments need to be conducted.**
> >
> > **Re:**
> > Thank you very much for your suggestions; we will be based on your suggestions to further improve our work. Our current work does not focus on the impact of DreamerV3's hyperparameters on the algorithm for the following reasons:
> > 1. DreamerV3 is a general algorithm that learns to master diverse domains while using fixed hyperparameters, making reinforcement learning readily applicable.
> > 2. The effects of different hyperparameters have been extensively studied [2, 3, 4].
> >
> > However, we are willing to perform ablation analyses of the effects of hyperparameters after the completion of the existing experiments if this can be done during rebuttal.
> >
> > ## Weakness 5
> > **Whether modeling two world models will lead to a significant increase in learning time and the effectiveness of model learning also need to be considered.**
> >
> > **Re:**
> > Thank you very much for your question; the suggestion of incorporating model efficiency is very valuable. Based on our comparisons, modeling a two-world model resulted in about a 40% increase in training elapsed time. We will be collating the results into a chart and updating it shortly.
> >
> > ## Weakness 6
> > **Privileged information and partial observations should be considered in more environments. A single scenario in QuadrotorGoal1 cannot be trusted to determine whether privileged information enhances performance.**
> >
> > **Re:**
> > Thank you very much for your suggestion. We have introduced a new environment, SafetyRacecarGoal1, to conduct ablation experiments and will provide additional experimental data to analyze the impact of privileged information. The results of these experiments will be updated shortly.
> >
> > [2] Ma, H., Wu, J., Feng, N., Xiao, C., Li, D., Hao, J., Wang, J., and Long, M. (2024). Harmonydream: Task harmonization inside world models.
> >
> > [3] Hafner, D., Pasukonis, J., Ba, J., and Lillicrap, T. (2024). Mastering diverse domains through world models.
> >
> > [4] Huang, W., Ji, J., Xia, C., Zhang, B., and Yang, Y. (2024). Safedreamer: Safe reinforcement learning with world models.

---

> ### Author Response · Authors · 2024-11-22
> **Response to Reviewer xknH (3)**
>
> We have incorporated all changes into the latest version of the PDF. In response to your concerns, we have made the following revisions:
>
> - In the practical implementation section, we added an explanation of loss optimization, clarifying the distinctions between our approach and SafeDreamer regarding this aspect.
> - We included a baseline description in Appendix E.
> - We added a training time comparison in Appendix E, demonstrating that the incorporation of a world model resulted in an average increase of 50% in training time.
> - In response to Weakness 6, we conducted an ablation experiment in the RacecarGoal1 scenario. **Our experimental results indicate that the addition of privileged information leads to a significant performance increase in both the RacecarGoal1 and QuadrotorGoal1 scenarios**. Additionally, the bsrp_Lag and AsymDreamer models shown in Figure 4 serve as ablation experiments for the inclusion of privileged world models, illustrating that the integration of privileged information substantially enhances model performance. These empirical findings provide strong evidence for the advantages of incorporating privileged information.
>
> We hope the above revisions adequately address your concerns, and we look forward to your feedback.

---

### Official Review · Reviewer_28Vi · 2024-11-03

**Soundness:** 2
**Presentation:** 2
**Contribution:** 2
**Rating:** 5
**Confidence:** 4

**Summary:**

This paper focuses on the safe RL problem that struggles with performance degradation and often fails to satisfy safety constraints. They attribute this problem to the lack of necessary information in partial observations and inadequate sample efficiency. Specifically, they exploit low-dimensional privileged information to build world models, thereby enhancing the prediction capability of critics. The authors propose Asymmetric Constrained Partially Observable Markov Decision Processes, a relaxed variant of CPOMDPs. The key distinction is that ACPOMDPs assume the availability of the underlying states when computing the long-term expected values. To ensure safety, they employ the Lagrangian method to incorporate safety constraints. The experiments conducted on the SafetyGymnasium benchmark demonstrate that the proposed approach outperforms existing approaches dramatically in terms of performance and safety.

**Strengths:**

1.	The idea of using privileged information in the Dreamer structure is interesting. Separating observation modeling and task-centric prediction modeling avoids the potential trade-off between these two tasks. This also allows the observation world model to capture more detailed observation information, thus enabling the actor model to achieve better performance with richer input features.
2.	The paper is generally well-written and well-organized. Figure 1 clearly shows the training pipeline.

**Weaknesses:**

1.	**Justification of using privileged information**. It seems that the motivation for using privileged information is this sentence “Since training is often conducted in simulators, there is potential to leverage privileged information during training to reduce uncertainty from partial observations”. Does the proposed method have potential applications beyond simulations, such as in real-world scenarios? In addition, I am not sure if it is fair to compare with other methods that do not use privileged information.
2.	**Evaluation results**. The experimental evaluation is only conducted on 4 tasks, including one self-made task. The authors may need to include all the remaining tasks in the Safety Gymnasium.
3.	**Missing baseline on the same benchmark** Although this paper proposes a model-based method, I think it is still meaningful to compare with some well-known methods on the same benchmark. This website contains some results that can be used as reference: https://fsrl.readthedocs.io/en/latest/tutorials/benchmark.html.
4.	There is no clear evidence to show the benefit of using privileged information. The results in Figure 5 need more investigation and explanation. Otherwise, it is hard to summarize the main conclusion of the proposed method.

**Questions:**

1.	I think in Figure 1, the encoder, decoder, and hidden state are different for the two world models. So, it would be clearer to use different colors of notations for them to avoid confusion.
2.	Why not use SafeDreamer as a baseline?
3.	In Figure 4, why do all baselines have constant cost? Don’t they vary across training steps? What is the target cost limit?
4.	What does the red solid line mean in Figure 4 and Figure 5?
5.	There is no explanation of the baselines. What does OSRP mean? It would be better to include simple descriptions of each baselines in the appendix.
6.	The ablation study results in Figure 5 raise a lot of questions. I think the authors also feel surprised that the privileged world model fails to train a viable cost predictor when taking privileged information as input. In addition, DreamerV3 is much better than AsymDreamer(S) in terms of the reward. This is also hard to interpret. I think the authors should do more experiments to explain these results to provide insights into the model. Otherwise, there seems no clear message that can be summarized in this paper.

---

> ### Author Response · Authors · 2024-11-18
> **Response to Reviewer 28Vi (1)**
>
> Thank you for your thorough review and valuable suggestions regarding this paper. Your suggestions have greatly helped us to improve our paper. **The modifications can be seen in the new version of the pdf.** In response to your concerns, we have provided detailed replies, added more tasks to compare with the baseline algorithms, added more ablation studies. We hope these address your concerns and would appreciate it if you could adjust your scores accordingly!
>
> 1. **Question 1.** I think in Figure 1, the encoder, decoder, and hidden state are different for the two world models. So, it would be clearer to use different colors of notations for them to avoid confusion.
>
>    **Re:** Thank you very much for your advice. We have updated Figure 1 by adding superscripts to indicate components associated with different world models. The notation 'o' denotes components belonging to the observation world model, while 'p' signifies components associated with the privileged world model.
>
> 2. **Question 2.** Why not use SafeDreamer as a baseline?
>
>    **Re:** In fact, SafeDreamer is our main comparison. OSRP, OSRP_Lag, BSRP_Lag are three algorithms proposed by SafeDreamer respectively. **We have added a description of the baseline algorithms in the appendix F to avoid any confusion this may cause.**
>
> 3. **Question 3.** In Figure 4, why do all baselines have constant cost? Don’t they vary across training steps? What is the target cost limit?
>
>    **Re:** This is because plotting the cost curves for all algorithms would result in a cluttered figure. Therefore, for all BASELINE algorithms, we only use dashed lines to represent the average cost at the end of training. The target cost limits for all tasks are set at 2. **We have modified the figure captions to illustrate our experimental setup more clearly.**
>
> 4. **Question 4.** What does the red solid line mean in Figure 4 and Figure 5?
>
>    **Re:** The red solid line represents the target cost limit.
>
> 5. **Question 5.** There is no explanation of the baselines. What does OSRP mean? It would be better to include simple descriptions of each baseline in the appendix.
>
>    **Re:** OSRP is an algorithm proposed by SafeDreamer that integrates Dreamerv3 with CCEM for decision time planning. **We have added a description of the baseline algorithms in the appendix F.**

---

> ### Author Response · Authors · 2024-11-18
> **Response to Reviewer 28Vi (2)**
>
> 6. **Question 6.** The ablation study results in Figure 5 raise a lot of questions. I think the authors also feel surprised that the privileged world model fails to train a viable cost predictor when taking privileged information as input. In addition, DreamerV3 is much better than AsymDreamer(S) in terms of the reward. This is also hard to interpret. I think the authors should do more experiments to explain these results to provide insights into the model. Otherwise, there seems no clear message that can be summarized in this paper.
>
>    **Re:** Thank you very much for your suggestions! Additional ablation experiments have been conducted to address the issue of privileged world models occasionally failing to train effective cost predictors.
>
>    DreamerV3 achieves higher rewards than AsymDreamer(S) for two primary reasons:
>    1. **DreamerV3 is not designed with safety in mind.** It does not incorporate safe decoders or safe critics, allowing actors to pursue higher rewards without regard for potential dangers. In contrast, AsymDreamer(S) considers the avoidance of hazardous behaviors, resulting in inherently lower rewards compared to DreamerV3.
>    2. **Relying solely on privileged information does not yield the desired outcomes.** Research[1] has shown that monitoring based exclusively on privileged information can lead to suboptimal performance, as it may cause actors to focus solely on maximizing rewards at the expense of observing their behavior in the environment.
>
>    Therefore, our model for AsymDreamer integrates both observational information and privileged information for supervision. **This approach mitigates the risk of neglecting environmental observations by the actors and ultimately leads to significantly improved outcomes.**
>
>    We are conducting more ablation experiments and collating experimental data to better illustrate the benefits of using privileged information. **More details of the experiments and the updated results will be presented soon.**
>
> [1] Karkus, P., Hsu, D., and Lee, W. S. (2017). Qmdp-net: Deep learning for planning under partial
> observability.

---

> ### Author Response · Authors · 2024-11-19
> **Response to Reviewer 28Vi (3)**
>
> 1. **Weakness 1.** Justification of using privileged information. It seems that the motivation for using privileged information is this sentence “Since training is often conducted in simulators, there is potential to leverage privileged information during training to reduce uncertainty from partial observations”. Does the proposed method have potential applications beyond simulations, such as in real-world scenarios? In addition, I am not sure if it is fair to compare with other methods that do not use privileged information.
>
>    **Re:**
>    Thank you for bringing this issue to our attention. It is noteworthy that the integration of privileged information in reinforcement learning has been thoroughly studied and successfully applied in real-world scenarios within the field of robotics [1,2,3,4]. These methodologies typically involve training algorithms with privileged information in a simulated environment, followed by their deployment in real-world settings through the Sim2Real approach.
>
>    Reference [2] implemented real-world deployment using an asymmetric actor-critic algorithm and domain randomization, while reference [3] achieved real-world deployment through the Dreamer framework and domain randomization. These approaches utilize similar architectures to our work, and thus, the success of their real-environment deployment provides a strong assurance for the real-world deployment of our algorithms. Consequently, we believe that our algorithm can be applied to real-world scenarios.
>
>     **Question: In addition, I am not sure if it is fair to compare with other methods that do not use privileged information.**
>
>     Thank you for your valuable question. We consider safe model-based reinforcement learning algorithms to be our main comparison. To the best of our knowledge, we are the first to use privileged information in the field of safe model-based reinforcement learning, which makes it challenging to identify a baseline that also uses privileged information.
>
>     While some existing approaches, such as TWIST, Scaffolder, and Informed Dreamer, have attempted to integrate privileged information into model-based reinforcement learning. But none of these frameworks consider safe constraints. Incorporating safe constraints in these approaches and comparing them with our framework will be our subsequent research direction.
>
> 2. **Weakness 2.** Evaluation results. The experimental evaluation is only conducted on 4 tasks, including one self-made task. The authors may need to include all the remaining tasks in the Safety Gymnasium.
>
>    **Re:**
>    We are currently expanding our range of environments to facilitate comparisons with baselines. In previous studies, LAMBDA and Safe-SLAC were tested in six environments, while SafeDreamer was evaluated in five. To enrich our experiments, we have added two new environments: *SafetyCarGoal1* and *SafetyRacecarGoal1*. We opted not to include the *SafetyPointGoal1* task, as it is too simplistic to adequately reflect the differences among the various algorithms.
>
>    We will update the results of our experiments shortly, and we anticipate that the forthcoming details and findings will enhance your understanding of the effects of incorporating privileged information.
>
> 3. **Weakness 3.** Missing baseline on the same benchmark. Although this paper proposes a model-based method, I think it is still meaningful to compare with some well-known methods on the same benchmark. This website contains some results that can be used as reference: [https://fsrl.readthedocs.io/en/latest/tutorials/benchmark.html](https://fsrl.readthedocs.io/en/latest/tutorials/benchmark.html).
>
>    **Re:**
>    It may be challenging for us to include these baseline experiments in the limited time available during the defence. However, we are more than happy to conduct comparative analyses using the results from the website you have provided. We are currently collating the results and will be updating them shortly.
>
> 4. **Weakness 4.** There is no clear evidence to show the benefit of using privileged information. The results in Figure 5 need more investigation and explanation. Otherwise, it is hard to summarize the main conclusion of the proposed method.
>
>    **Re:**
>    We are incorporating additional ablation experiments to analyze the implications of privileged information.
>
> [1] Miki, T., Lee, J., Hwangbo, J., Wellhausen, L., Koltun, V., and Hutter, M. (2022). Learning robust perceptive locomotion for quadrupedal robots in the wild. *Science Robotics*, 7(62).
>
> [2] Pinto, L., Andrychowicz, M., Welinder, P., Zaremba, W., and Abbeel, P. (2017). Asymmetric actor critic for image-based robot learning.
>
> [3] Yamada, J., Rigter, M., Collins, J., and Posner, I. (2023). Twist: Teacher-student world model distillation for efficient sim-to-real transfer.
>
> [4] Lee, J., Hwangbo, J., Wellhausen, L., Koltun, V., and Hutter, M. (2020). Learning quadrupedal locomotion over challenging terrain. *Science Robotics*, 5(47).

---

> ### Author Response · Authors · 2024-11-22
> **Response to Reviewer 28Vi (4)**
>
> We have incorporated all changes into the latest version of the PDF. In response to your concerns, we have made the following revisions:
>
> - Optimized the display of Figure 1 to enhance the clarity of the differences between the two world models.
> - Provided a more detailed explanation of Figures 4 and 5.
> - Included a description of the baselines in Appendix E.
> - For Weakness 3, we added a comparative experiment with the model-free algorithm in Appendix E. **The experimental results demonstrate that AsymDreamer significantly outperforms other algorithms in terms of security and reward.**
> - For Weakness 2, we included two sets of comparative experiments between RacecarGoal1 and CarGoal1.
> - In response to Question 4, we added an ablation experiment in RacecarGoal1. Our findings indicate that it is feasible to train a usable cost decoder using privileged information within the RacecarGoal1 scenario.
> The primary reasons for the suboptimal performance of privileged information in QuadrotorGoal1 are:
>   1. the cost distribution is extremely imbalanced.
>   2. privileged information requires the learning of additional information to effectively model the cost function.
>
> We hope the above revisions address your concerns, and we look forward to your feedback.

---

### Official Review · Reviewer_sZ2A · 2024-11-03

**Soundness:** 2
**Presentation:** 3
**Contribution:** 2
**Rating:** 5
**Confidence:** 4

**Summary:**

The authors propose AsymDreamer, a Dreamer-based safe reinforcement learning framework that utilizes low-dimensional privileged information to construct world models. The world model in AsymDreamer features two branches: the Privileged World Model, which takes a handcrafted low-dimensional vector as input, and the Observation World Model, which uses partially observed images (64x64 RGB). Additionally, the authors formulate their approach within the framework of Asymmetric CPOMDPs (ACPOMDPs) and integrate AsymDreamer with the Lagrangian method. Empirical results show that AsymDreamer outperforms existing safe reinforcement learning methods on the Safety-Gymnasium benchmark.

**Strengths:**

1. The theoretical analysis of ACPOMDPs in this paper is comprehensive. The authors propose the ACPOMDPs framework and provide a detailed analysis showing that asymmetric inputs reduce the number of critic updates and lead to a more optimal policy compared to the standard CPOMDPs framework.
2. The authors introduce privileged information into the RSSM world model, enhancing the model's imaging capabilities and consequently improving the performance of the safe policy.

**Weaknesses:**

1. The methodology in this paper is too similar to prior work[1], and lacks sufficient novelty. In my view, the only difference is the inclusion of privileged information in the modeling of the RSSM world model.
2. The authors are encouraged to include an ablation study to analyze the impact of adding privileged information to the observation input on the baseline algorithm.
3. There are some typos in the paper that need to be corrected in the next version (e.g., line 400 and figure 2).

[1] Huang, Weidong, et al. "Safe dreamerv3: Safe reinforcement learning with world models." arXiv preprint arXiv:2307.07176 (2023).

**Questions:**

1. Please clarify the similarities and differences between the proposed method and SafeDreamer[1].
2. The reviewer is confused about what the global state in your privileged world model input is and where the privileged information (the low-dimensional vector from Appendix E) is used.

[1] Huang, Weidong, et al. "Safe dreamerv3: Safe reinforcement learning with world models." arXiv preprint arXiv:2307.07176 (2023).

---

> ### Author Response · Authors · 2024-11-18
> **Response to Reviewer sZ2A （1）**
>
> Thank you for your thorough review and valuable suggestions regarding this paper. Your comments and concerns will significantly contribute to enhancing our work. **The modifications can be seen in the new version of the PDF.** In response to your concerns, we have provided detailed replies, revised the methodology section, and added more ablation studies. We hope these changes address your concerns, and we would appreciate it if you could adjust your scores accordingly!
>
> 1. **Weakness 1 & Question 1:** Please clarify the similarities and differences between the proposed method and SafeDreamer [1].
>
>    **Re:** Thank you very much for pointing this out! We have rewritten our methodology into two sections, METHODS and PRACTICAL IMPLEMENTATION, to emphasize our key contributions.
>
>    Our primary contribution is the proposal of an asymmetric training structure that effectively utilizes privileged information in model-based reinforcement learning. This approach offers four distinct benefits:
>    - It allows us to avoid potential trade-offs between observational modeling and task-centric predictive modeling.
>    - The asymmetric training structure enables the privileged world model to access all information from the observational world model, thereby enhancing its reward decoder and cost decoder.
>    - This structure facilitates both the observed world model and the privileged world model in capturing the maximum amount of information.
>    - The asymmetric training structure can be seamlessly transferred to implementations of other world models.
>
>    The theoretical analysis of ACPOMDP is also a significant aspect of our work. Our comparative analysis with CPOMDP demonstrates that substantial improvements can be achieved by employing an asymmetric structure:
>    - It reduces the number of updates required by the critic to estimate the value function.
>    - It leads to more optimal policies compared to the standard CPOMDP framework.
>
>    We implemented our method using RSSM and Augmented Lagrangian methods primarily because both approaches are highly regarded and widely accepted in the safety reinforcement learning community, supported by a substantial body of research. **However, we would like to emphasize that our approach is not dependent on these specific techniques.** In fact, our method can be easily adapted to Bayesian world models, Transformer-based world models, and latent variable world models. Furthermore, the augmented Lagrangian method can be entirely replaced with alternative methods, such as LBSGD.
>
>    SafeDreamer’s work can be divided into two main parts:
>    - BSRP_Lag: Policies optimization by integrating Dreamerv3 with Lagrangian methods.
>    - OSRP and OSRP_Lag: Decision-time planning through the combination of Dreamerv3 with the CCEM.
>
>    We regard OSRP as the primary contribution of this work for the following reasons: The integration of world models with Lagrangian methods, as seen in BSRP_Lag, has been extensively studied, including approaches like LAMBDA, which combines Bayesian world models with Lagrangian methods, and Safe-SLAC, which merges latent variable world models with Lagrangian methods. Therefore, we consider the contribution of BSRP_Lag's implementation of world models using Dreamerv3 to be minimal and lacking in innovation. Ultimately, we will summarize the similarities and differences between our work and SafeDreamer.
>
>    **Similarities:**
>    - Both approaches utilize RSSM and augmented Lagrangian methods for implementation.
>
>    **Differences:**
>    - Our work emphasizes leveraging privileged information to enhance the effectiveness of model-based safe reinforcement learning algorithms, whereas SafeDreamer focuses on utilizing world models to achieve zero-cost performance.
>    - Our approach is presented in the CPOMDP context, which emphasizes partial observability, while SafeDreamer is built within the more generalized CMDP framework.
>    - SafeDreamer's primary focus is decision-time planning using CCEM, a method that is hindered by time-consuming execution. In contrast, our work leverages privileged information to optimize a more effective policy, thereby avoiding prolonged execution times.
>    - In our approach, both RSSM and augmented Lagrangian methods can be freely replaced without impacting our core contributions (asymmetric training structure and ACPOMDPs). In contrast, SafeDreamer, as a specific algorithm, is entirely dependent on RSSM and augmented Lagrangian methods.
>    - In addition to empirical research, we formalize and theoretically analyze the use of privileged information in model-based reinforcement learning algorithms as ACPOMDPs. SafeDreamer, on the other hand, is solely an empirical study and does not include a theoretical analysis.

---

> ### Author Response · Authors · 2024-11-18
> **Response to Reviewer sZ2A （2）**
>
> - **Weakness 2:** The authors are encouraged to include an ablation study to analyze the impact of adding privileged information to the observation input on the baseline algorithm.
>
>    **Re:** Thank you for your feedback. It is unclear whether you require direct training with low-dimensional privileged information in the baseline algorithms (LAMBDA, Safe-SLAC) or if you expect us to implement our asymmetric training structure within these baseline algorithms.
>
>    In the former case, we can attempt to provide experimental results before the conclusion of the rebuttal period. However, in the latter case, implementing the asymmetric training structure in the baseline would necessitate significant code modifications, which may present challenges during the rebuttal.
>
>    We are also in the process of adding more environments (SafetyCarGoal1, SafetyRacecarGoal1) to facilitate comparisons with the baseline algorithms and to conduct ablation experiments across these additional environments. We will update the results of our experiments shortly, and we hope that the forthcoming experimental details and results will enhance your understanding of the impact of incorporating privileged information.
>
> - **Question 2:** The reviewer is confused about what the global state in your privileged world model input is and where the privileged information (the low-dimensional vector from Appendix E) is used.
>
>    **Re:** We apologize for the lack of clarity in our manuscript. In fact, the privileged information (the low-dimensional vector from Appendix E) is the global state. We have modified our manuscript to uniformly replace all instances of *global state* with *privileged information* to avoid ambiguity, which can be seen in the Methods section.
>
> - **Weakness 3:** There are some typos in the paper that need to be corrected in the next version (e.g., line 400 and figure 2).
>
>    **Re:** We have corrected these errors and have meticulously reviewed the rest of the manuscript for any additional mistakes.
>
>  Please let me know if you would like any further adjustments!

---

> ### Author Response · Authors · 2024-11-22
> **Response to Reviewer sZ2A （3)**
>
> We have incorporated all changes into the latest version of the PDF. In response to your concerns, we have made the following revisions:
>
> - We have corrected all errors in the manuscript.
> - In addressing Question 1, we have rewritten the methodology section to emphasize our core contribution and to clarify the similarities and differences with SafeDreamer.
> - We conducted a new ablation experiment in the RacecarGoal1 scenario to investigate the impact of privileged information. **Our results indicate that privileged information significantly enhances the baseline algorithm's ability to achieve rewards.**
>
> We hope the above revisions adequately address your concerns, and we look forward to your feedback.

---

### Official Review · Reviewer_A1zn · 2024-11-05

**Soundness:** 3
**Presentation:** 3
**Contribution:** 3
**Rating:** 6
**Confidence:** 2

**Summary:**

This paper presents AsymDreamer, an approach based on the Dreamer framework that specializes in exploiting low-dimensional privileged information to build world models, thereby enhancing the prediction capability of critics. AsymDreamer employs the Lagrangian method to incorporate safety constraints. This paper formulates the proposed approach as an Asymmetric CPOMDPs (ACPOMDPs) framework. Experiments on the Safety-Gymnasium benchmark demonstrate that AsymDreamer outperforms existing approaches in both performance and safety.

**Strengths:**

- The paper is well-motivated and novel. It successfully extends the Dreamer framework to handle asymmetric information in RL, presenting a novel ACPOMDP framework. It shows the effectiveness of using privileged Information.
- The results from Safety-Gymnasium benchmarks show that AsymDreamer outperforms baseline models in both task performance and safety metrics, especially in complex scenarios like 3D navigation. The thorough ablation studies are also conducted.
- The paper includes rigorous theoretical analysis and compares ACPOMDP to standard CPOMDP.

**Weaknesses:**

I am not in this field so I am unable to evaluate the signficance of the proposed method and ACPOMDP problem/analysis.

Some of my concerns include:
- Over-reliance on privileged information:  AsymDreamer’s performance relies on privileged information during training, which may not be available or hard to simualte in real-world environments. It potentially leads to bad performance or compromised safety in environments with limited or unavailable privileged information.

- Extension to real-world environments:  The experiments on Safety-Gymnasium benchmark are a bit toyish (even the most challenging 3D navigation one, although it may also be the cases for the related work). Additional testing in real-world environments would be beneficial to demonstrate the effectiveness.

**Questions:**

I am not familar with this field at all and I cannot evaluate the novelty of this work (especially over its predecessor / related works). It is an interesting read and I didn't identify significant issues. The paper is well motivated with solid analysis and experiments. My major concern is that this paper only evaluates on toyish environments and may not generalize well to real-world scenarios.

---

> ### Author Response · Authors · 2024-11-17
> **Response to Reviewer A1zn**
>
> Thank you for your thorough review and valuable suggestions regarding this paper. Your comments and concerns will significantly contribute to enhancing our paper. In response to your concerns, we have provided detailed replies.
>
> 1. **Weakness 1.** Over-reliance on privileged information: AsymDreamer’s performance relies on privileged information during training, which may not be available or hard to simulate in real-world environments. It potentially leads to bad performance or compromised safety in environments with limited or unavailable privileged information.
>
>    **Re:**
>    Thank you for bringing this issue to our attention. It is noteworthy that the integration of privileged information in reinforcement learning has been thoroughly studied and successfully applied in real-world scenarios within the field of robotics [1,2,3,4]. These methodologies typically involve training algorithms with privileged information in a simulated environment, followed by their deployment in real-world settings through the Sim2Real approach.
>
>    Reference [2] implemented real-world deployment using an asymmetric actor-critic algorithm and domain randomization, while reference [3] achieved real-world deployment through the Dreamer framework and domain randomization. These approaches utilize similar architectures to our work, and thus, the success of their real-environment deployment provides a strong assurance for the real-world deployment of our algorithms. Consequently, we believe that the challenge of acquiring accurate privileged information is not a significant limitation for our algorithm.
>
> 2. **Weakness 2.** Extension to real-world environments: The experiments on Safety-Gymnasium benchmark are a bit toyish (even the most challenging 3D navigation one, although it may also be the cases for the related work). Additional testing in real-world environments would be beneficial to demonstrate the effectiveness.
>
>    **Re:**
>    We are sorry to say that, due to limited resources, we are unable to conduct experiments in the real world. However, the Safety-Gymnasium benchmark is a very popular benchmark in the Safety Reinforcement Learning community, and the vast majority of the related works has been conducted in that benchmark rather than in the real world. Therefore, it is also sufficient to conduct experiments in that environment to illustrate the efficiency.
>
> Please let me know if you would like any further adjustments!
>
> [1] Miki, T., Lee, J., Hwangbo, J., Wellhausen, L., Koltun, V., and Hutter, M. (2022). Learning robust perceptive locomotion for quadrupedal robots in the wild. *Science Robotics*, 7(62).
>
> [2] Pinto, L., Andrychowicz, M., Welinder, P., Zaremba, W., and Abbeel, P. (2017). Asymmetric actor critic for image-based robot learning.
>
> [3] Yamada, J., Rigter, M., Collins, J., and Posner, I. (2023). Twist: Teacher-student world model distillation for efficient sim-to-real transfer.
>
> [4] Lee, J., Hwangbo, J., Wellhausen, L., Koltun, V., and Hutter, M. (2020). Learning quadrupedal locomotion over challenging terrain. *Science Robotics*, 5(47).

---

> > ### Comment · Reviewer_A1zn · 2024-11-22
> > **Response to Authors**
> >
> > Thanks for the detailed replies. I have no other questions and my concerns are addressed. From other reviewers, the major concerns are novelty over existing works (e.g., SafeDreamer) and experiments. The authors have provided discussions and additional experiments in the revision. I would like to keep the orginal score.

---

### Author Response · Authors · 2024-12-01
**Global Response to Reviewers**

We sincerely thank the reviewers for their detailed and insightful comments. Based on the feedback received, we have made the following revisions to the manuscript:
- We corrected errors throughout the manuscript and revised the methodology section to clearly highlight the differences from SafeDreamer.
- We added the SafetyRacecarGoal1 and SafetyCarGoal1 scenarios to enhance the comparison experiments.
- We conducted an ablation experiment in the SafetyRacecarGoal1 scenario to provide a clearer explanation and investigation of the role of privileged information.
- We included a comparison experiment with model-free baselines, demonstrating that AsymDreamer is significantly more effective than the baselines.
- We performed a comparative analysis of the training elapsed time for different algorithms, revealing that the inclusion of privileged information results in a 50% increase in training elapsed time.

The reviewers' comments have been instrumental in improving our paper. We kindly request that the reviewers review our responses and reconsider their scores.

---

### Meta-Review · Area_Chair_ZKVr · 2024-12-19

**Metareview:**

This paper approaches the problem of safe reinforcement learning in partially-observed environments. The paper proposes a method that centers around a particularly type of constrained POMDP that enables privileged access to state observations in certain settings. Reviewers found this paper well-motivated and novel, although doubts were raised about the sufficiency of the experimental comparisons, the justification for using privileged information potentially limiting the application to simulation only, and that there's no clear evidence to show the benefit of using privileged information.

Authors responded to these weaknesses in a variety of ways. Despite encouragements to engage, all but 1 reviewer did not meaningfully engage with the author responses (the reviewer who rated this paper a 6). They were the only reviewer rated this paper above the acceptance threshold, albeit with low confidence (2).

Given the limited engagement from reviewers, the only very-mildly positive initial reception, and my own read of the paper to check for things that the reviewers missed, e.g., if the paper was highly novel in a way that reviewers misunderstood, I recommend rejection. I concur with the overall assessment that more experimental validation and written justification is needed to clarify and support the paper's claims, particularly in providing evidence of the applicability of the setting / assumptions to solve non-toy problems. I also think the paper could be improved with an overview figure that makes clear the distinction between when privileged information is available and when it is not, and ground it in an example (or examples) from the experiments.

**Additional Comments On Reviewer Discussion:**

See the above main body of the metareview -- the discussion was quite limited despite encouragements to engage. Either the author responses failed to compel the reviewers, or the reviewers failed to engage, or a mixture of those factors occurred. In either case, there's not currently sufficient evidence to recommend acceptance.

---

### Decision · Program_Chairs · 2025-01-22

Reject